# Individualized Breakfast Programs or Glycogen Super-Compensation: Which Is the Better Performing Strategy? Insights from an Italian Soccer Referees Cohort

**DOI:** 10.3390/ijerph17031014

**Published:** 2020-02-05

**Authors:** Rebecca Regnoli, Marco Rovelli, Vincenzo Gianturco, Fabrizio Ernesto Pregliasco, Bruno Dino Bodini, Luigi Gianturco

**Affiliations:** 1IRCCS Galeazzi Orthopedic Institute, Dietician Service, 20161 Milan, Italy; rebecca.regnoli@gmail.com; 2Biomedical Committee Italian Soccer Referees’ Association (AIA-FIGC), 20100 Milan, Italy; marco.rovelli@icloud.com; 3Geriatrics Operative Unit, INRCA-IRCCS, 63900 Fermo, Italy; vincenzo.gianturco@gmail.com; 4IRCCS Galeazzi Orthopedic Institute, Department of Biomedical Sciences for Health, University of Milan, 20161 Milan, Italy; fabrizio.pregliasco@grupposandonato.it; 5Pulmonology Unit, Casati Hospital, 20017 Passirana, Italy; bodinibruno@hotmail.com

**Keywords:** referee, YYiR1, glycogen supercompensation, breakfast, diet

## Abstract

The role of soccer referees has grown in importance in the last decades, as has attention to their performance, which may be influenced and improved with specific and evolved training programs. Today, multiple specialists are working as a team in order to develop effective training programs. Moreover, for athletes, it is becoming more and more important to be attentive to nutrition. By considering such items, in this study, we aimed to investigate the nutritional habits of a group of referees belonging to the Italian Soccer Referees’ Association (on behalf of AIA-FIGC). Our main aim was to spread a “culture of nutrition” in refereeing, starting with a survey on referees’ breakfast attitudes and in order to disseminate such a “culture”, we chose top-level elite referees who were younger subjects (despite the average 4 years’ experience). Therefore, we enrolled 31 subjects (aged 22.74 ± 1.79, BMI 22.30 ± 1.53) and asked them about their breakfast habits. Then, for measuring their performance, we used the conventional fitness test named Yo-Yo (YYiR1), performed in three different sessions (test 1, test 2, test 3). Test 1 was carried out without any nutritional indications, test 2 was given after individualized breakfast suggestions by a designed dietician, and test 3 after an individualized glycogen super-compensation strategy. The Wilcoxon statistical analysis indicates that following an individualized breakfast strategy may enhance referees’ performance (*p* < 0.0001), whereas no significant effects were observed with the glycogen super-compensation option. However, further studies will be necessary to better address this topic and clarify whether high-carbohydrates (high-CHO) intake may be useful in other sports.

## 1. Introduction

Today, soccer referees are considered *out-and-out* athletes because of the evolution of the sport [1]. Thus, leading soccer federations have developed systems for managing referees’ performances and health conditions [2].

In the literature, several studies about soccer referees can be found. In particular, many studies have attempted to measure their level of performance by means of VO2 maximal uptake [3,4,5] and/or results of fitness field tests [6] and laboratory test [7,8].

Over the years, findings obtained in research applied to soccer refereeing have allowed consistent improvements in terms of training methods and/or types of tests evaluating fitness and/or specific physical skills [9,10].

Today, the main tool to assess the fitness status of referees is the Yo-Yo intermittent recovery test (YYiR1) [11].

Recently, Schenk et al. [12] have also analyzed the exercise physiology and nutritional perspectives of elite soccer referees since nutrition is more and more pivotal for individuals in sport. Their study directs special attention on education about adequate nutrition, fluid intake, dietary prevention of deficiencies (iron in female referees, vitamin D irrespective of sex and age), and basic precautions for travels abroad. The authors obviously pointed out the importance of daily carbohydrates (CHO) intake and affirmed that the simple adoption of nutritional considerations from active soccer for referees may not be appropriate. Recommendations should respect gender differences, population-specific physical characteristics, and demands just as much as individual characteristics and special needs. Moreover, the so-called FIFA (Federation International Football Association) 11+ prevention program has been tracing, since 2012, nutrition guidelines for soccer players [13], whilst there are no such publications for referees. Although referees and soccer players often differ in age, aerobic capacity and training load, nutritional choices could be crucial in for them order to achieve their best performance [14,15].

Therefore, we projected a study which may start a nutritional investigation of Italian soccer referees and attempt the glycogen super-compensation strategy. We chose that method because we considered it innovative and at the same time, easily applicable to the world of referees.

The glycogen super-compensation effect (achieving supraphysiological glycogen levels due to CHO depletion followed by loading) was first demonstrated in 1967 [16]. It occurs only when a low CHO diet is combined with vigorous exercise followed by a high-CHO diet. Since 1967, it has become a very popular pre-event performance enhancing strategy, especially for endurance athletes who benefit from glycogen super-compensation because fatigue in events lasting longer than one hour is related primarily to glycogen depletion. Although soccer referees are not endurance performers, it is important to evaluate this solution, even in such a particular type of athlete.

## 2. Methods

### 2.1. Participants

We enrolled 31 referees (mean age 22.74 ± 1.79) belonging to the Italian Soccer Referees Association (AIA) between October 2017 and April 2018. All of them had no cardiac history and the physical assessments followed the Italian eligibility criteria for practicing a competitive sport [17]. All the officials who volunteered to the protocol had a median of 4 years of refereeing experience.

The enrolled subjects did not use any supplement and/or ergogenic aids before or during testing.

Before starting the study, written informed consent was collected and the study protocol was approved by the Local Internal Ethics Committee of AIA (Protocol Number 001LG_CRLomb/1.2017). This study also did not provide any pharmacological prescriptions and/or invasive procedures.

### 2.2. Study Protocol

The protocol was performed in accordance with the Declaration of Helsinki [18] and met the ethical standards for Sport and Exercise Science Research.

The timeline of the study contemplated three different assessments (Table 1: t1, t2, t3). The intervals between the tests was about 45 days. In each session, physicians documented weight and BMI and then registered the YYiR1 results.

The first investigation provided dietary indications before YYiR1 but researchers asked them about breakfast attitude by means of a simple food diary. Thus, in t2, physicians required that participants consumed an individualized specific breakfast in the morning of the fitness test based on characteristics deduced by answers in their food diaries. Finally, before t3, performance biomedical staff of AIA prescribed to athletes an individual diet program based on glycogen super-compensation. That program was suggested to be left from the 7th day before the YYiR1 in agreement with the known prescriptions in the literature [19].

### 2.3. Glycogen Super-Compensation

There are many different ways to obtain glycogen super-compensation [20,21]. In our protocol, diet and exercise regimen started with a glycogen-depleting exercise bout. The exercise was then followed by 3 days of a high-protein, high-fat diet. Another exhausting exercise bout was performed on day 4, after which the subjects were placed on a high-CHO diet for 3 days. Finally, on day 7, enrolled athletes (soccer referees) underwent the YYiR1 test.

### 2.4. Yo-Yo Test

The YYiR1 test (also known as the Yo-Yo test) is the current most valuable strategy to measure specific fitness in soccer referees. It consists of 2 × 20 m (mt) runs back and forth between two lines at a progressively increasing speed controlled by audio inputs. When the referees fail twice to reach the corresponding line in time, the test finishes and the distance covered is recorded as the personal result. The minimum level of coverage is normally set at 2800 mt for Italian referees.

Each testing session was performed with well rested subjects, as has been done in the recent past [22]. In more detail, referees kept up moderate exercise during 24–48 h preceding YYiR1 (i.e., 20–30 min running at 70% of individual maximal heart rate). The referees performed the test on natural turf because of it was the prevalent type of pitch on which they worked.

Finally, the environmental conditions during the tests were similar for all the participants because of all the sessions were in the morning (about 1 h after breakfast) in analogue climatic conditions.

### 2.5. Statistical Analysis

The collected data were analyzed and statistics were computed with MedCalc software (Ostend, Belgium, version 12.7.0) by Schoonjans et al. [23]. Significance was conventionally set at 5% (*p* < 0.05).

Continuous variables were expressed as the mean ± standard deviation (SD) if normally distributed and/or as median and interquartile ranges (IQR) if skewed [24]. Categorical variables were identified as frequency (percentage).

Comparisons were performed using the paired-data Wilcoxon test (rank modality) by Fix and Hodges [25] and in order to detect possible outliers, we performed a *post-hoc* Tukey HSD (honestly significance differences) test [26].

## 3. Results

The starting assessments found a total average energy intake of 3030.30 ± 225.67 divided into CHO (49.76 ± 30.53%), proteins (11.09 ± 9.78%), and fats (9.78 ± 8.00%).

The Wilcoxon analysis revealed that the second session of the fitness test found an improvement in covered distance for YYiR1 (see Figure 1 for details) and also of VO2max if we use the Bangsbo formula (VO2max = distance in mt × 0.0084 + 36.4) for indirect calculation of maximal aerobic capacity [11]. There was, however, no significant difference between the second and third session.

Finally, the *post-hoc* Tukey analysis did not detect any outliers in our population (see Figure 2).

## 4. Discussion

Nutrition is a key factor in health promotion and a pivotal parameter for sports performance [27]. Therefore, today, nutritional strategies are a part of physical conditioning and training in athletes. By now, referees are considered as real athletes since they follow regular and intensive training programs to reach a high level of performance [28]. Generally, among referees, the estimation of performance is calculated by means of fitness tests, such as YYiR1, and in soccer, refereeing is partly related to endurance efforts with low-to-high intensity phases of running. Moreover, historically, the exercise models of soccer referees have mainly been related to the aerobic pathway [29] and indeed, it has already been shown that referees can run up to 10 km during a match [30,31]. YYiR1 is the main fitness test for all the Italian Soccer Referees (both elite and younger subjects).

The growing field of soccer-related scientific research has directed relatively little attention to nutrition and the link between soccer referees’ performance and nutrition has not totally been addressed yet [32]. However, in both male and female soccer players, the majority of studies have demonstrated an inadequate nutritional intake [33,34], underlying the need to improve short and long-term adherence to diet choices. There is substantial agreement on the need to design and implement individualized intervention programs for soccer players [35]. Keeping to a balanced diet can play a fundamental role in maximizing the body’s efficiency, especially in young athletes [36]. There is, therefore, a need for sports nutrition counselling and/or education, which would help to improve athletes’ eating habits, as well as optimizing their sports training performance. Therefore, promotion of “diet knowledge” was our main stakeholder. Generally, knowledge is strictly related to prevention strategies and better risk profiles in medicine [37] but in sports medicine, it is not so easy to translate “nutrition knowledge” in practice [38] and it may be also more difficult to transfer good dietary habits in improved performance. Nevertheless, the positive effects of dietary habits was observed when the dietician was the primary nutrition information source for young athletes, without any gender differences [39]. Analogously, we chose a dietician as the official nutrition “voice” for our athletes. After making this choice, we searched for studies about nutrition in refereeing and we discovered that they generally consume a diet that does not have sufficient CHO calories [40].

Other authors have focused their attention exclusively on the caloric expenditure of refereeing [41,42], whereas the aim of our study was to promote a sort of nutritional learning for soccer referees. Obviously, in order to do that, we analyzed the food habits and nutrient intake of soccer players by considering the same sport type [43].

By evaluating the existing gap between CHO supply and demand, we chose glycogen super-compensation as the main referral option. Many authors have already investigated this strategy in competitive sportive men and/or women [44,45] and we knew that exercise intensity is an important factor determining excess post-exercise oxygen consumption (EPOC) [46].

A larger EPOC is usually observed after high-intensity supra-maximal effort (SE) compared to a moderate one, even when the total work performed is matched [47]. Regarding the main paths to prolong SE duration, there is some evidence suggesting that a high-CHO availability has an important role [48]. Athletes may reach a higher exercise tolerance by choosing a high-CHO diet [49].

Returning to our population, high-CHO intake demonstrated a neutral effect on performance and even better stats revealed a worse tendency of YYiR1 results. These findings may be partly ascribed to the not-so-vast sample size and/or because refereeing is an atypical sport in which endurance is an important though not exclusive item. On the contrary, by implementing breakfast habits, referees obtained a better performance in the fitness test (t2 vs. t1, *p* < 0.0001). Therefore, the final message seems to be in line with the International Olympic Committee (IOC) consensus statement about diet, supplementation and high-performance athletes. In this document, IOC recommends a “healthy diet” and/or good diet habits as the first step to reaching the best performance [50].

## 5. Conclusions

In summary, our study confirms the importance of a correct pre-work-out meal for practicing sport and obtaining the best results. In fact, individualized breakfasts proved to be the winning solution.

A high CHO intake seems does not seem to be completely effective for improving soccer referees’ performance. Moreover, it is pivotal to encourage soccer referees to pay attention to their dietary choices by starting with dietician counselling and individualized breakfast programs, as already demonstrated in other athlete categories [51]. Obviously, our sample did not have a high amplitude and further studies are necessary in order to clarify the diet strategy able to improve soccer referees’ performance indirectly measured by means of a typical YYiR1 fitness test. For example, we could conduct similar studies with more enrolled subjects and measure performance by means of cardiopulmonary exercise testing (CPET) instead of with YYiR1.

## Figures and Tables

**Figure 1 ijerph-17-01014-f001:**
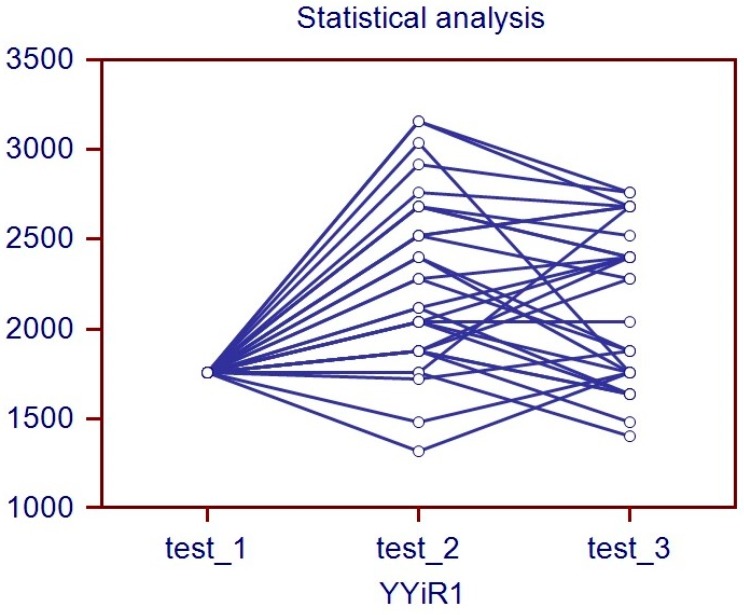
Statistical analysis. Wilcoxon test results → Meters obtained at YYiR1.

**Figure 2 ijerph-17-01014-f002:**
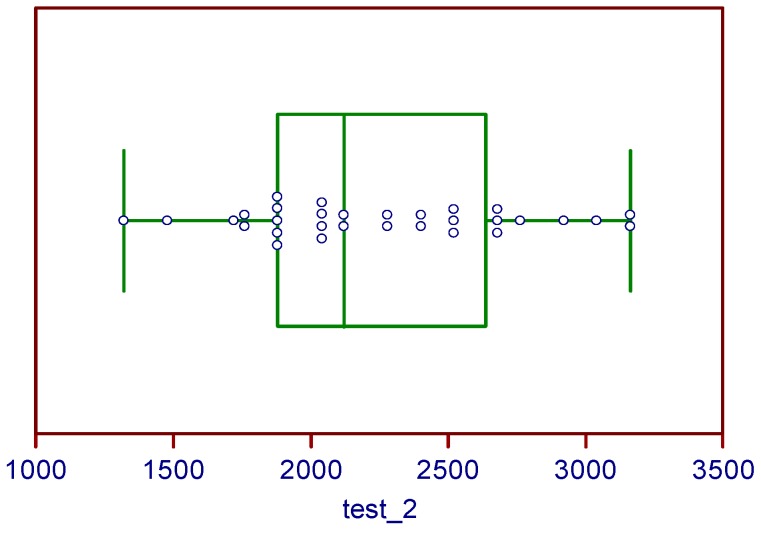
Outliers detection. Box-and-whisker plots for Tukey HSD (*post-hoc*) test: anyone outliers both in the 2nd and 3rd session of YYiR1.

**Table 1 ijerph-17-01014-t001:** Baseline patients’ characteristics and main results.

Items	*N* = 31
Age (years)	22.74 ± 1.79
Height (cm)	1.82 ± 0.06
Weight (kg)	74.05 ± 6.66
Gender male (%)	96.77
BMI (kg/m^2^)	22.30 ± 1.53
YYiR1	
Test 1 (m)	1760 (1760–1760)
Test 2 (m)	2120 (1880–2640) ^#^
Test 3 (m)	2280 (1760–2490) ^§^
Energy Intake (kcal)	2412.6 ± 566.6
Carbohydrates (%)	49.76 ± 30.53
Proteins (%)	11.09 ± 9.78
Fats (%)	9.78 ± 8.00
Refereeing Experience (age)	4 (2.8–6.2)

BMI = body mass index; YYiR1 = Yo-Yo Intermittent Recovery Test-type 1. Continuous variables were expressed as mean ± SD if normally distributed and/or as median and interquartile ranges (IQR) if skewed. Categorical variables were expressed as frequency (percentage). Comparisons performed using Wilcoxon (rank modality) test. ^#^
*p <* 0.05 (Test 2 vs. Test 1), ^§^
*p* = 0.13 (Test 3 vs. Test 2).

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
