# Peer review of "Individualized Breakfast Programs or Glycogen Super-Compensation: Which Is the Better Performing Strategy? Insights from an Italian Soccer Referees Cohort"

_ijerph, 2020, doi:10.3390/ijerph17031014_

Round 1

Reviewer 1 Report

Line 31 : Endurance is not the average activity of referees. Their kind of exertion must be considered a “stop and go” exercise, so it can’t be compared to endurance exercise. Line 89-91: There are many different ways to obtain the glycogen supercompensation. It is better to specify that this is one of the methods (Jeukendrup and Gleeson Sport Nutrition Human Kinetics Champaign IL, 2005; Sherman W.M. et al., Effect of exercisa-diet manipulation on muscle glycogen and its subsequent utilization during performance, 1981, Int J Sports Med) Results: in my opinion the graphs are not clear; it could be more useful and explicit to group the distance of all athletes into the 3 different stages, with the standard deviation. Discussion: although it has already been confirmed that carbohydrates before exercise are the best choice, from a practical point of view, it is unthinkable that every weeks referees could make a “glycogen supercompensation”. Therefore, I suggest adding a sentence to emphasize the importance of a correct pre-work out meal, as the best solution.

Author Response

"Please see the attachment".

Reviewer 2 Report

Introduction:

Introduction needs to contain more information.

Line 45. Schenk, et al. showed important information on the topic that should be the introduction (ex. Nutritional recommendatitions). Mention some recommendations in the article.

The introduction has no justification on the super-compensation glycogen method was chosen for this study. I think it's important to talk about the super-compensation glycogen and the reason for the choice on the introduction and in the methods talk only about the protocols of the super-compensation glycogen method.

Methods:

Line 56 to 60. It is important to mention football referee did or not the use of a supplement and/or ergogenic aids before or during testing.

Why do the referees who participated in the study have a low average age? Articles with soccer referees have higher average ages.

Line 99. How long after breakfast was the test performed? What is the interval between tests? Put this information in the article.

Results:

Line 115. It was not clear whether these data refer to individualized breakfast, glycogen super-compensation or breakfast without any nutritional indications. I suggest a table with a comparison of the three types of breakfast (energy consumption, macronutrients, etc.) to present the differences between them.

Line 123. I suggest a table with the results (mean, standard deviation, etc.) to facilitate the readers' understanding of the article.

Discussion:

Line 169. What is the opinion of the authors for the method not having improved the performance of the referees? Does the scientific literature explain this fact?

Conclusions:

What are the authors' suggestions for the next studies? Do I consider it important to mention in the conclusion?

Author Response

"Please see the attachment".

Round 2

Reviewer 1 Report

Now is better. Thank you